# First Report of *bla*_NDM-1_ Bearing IncX3 Plasmid in Clinically Isolated ST11 *Klebsiella pneumoniae* from Pakistan

**DOI:** 10.3390/microorganisms9050951

**Published:** 2021-04-28

**Authors:** Hazrat Bilal, Gaojian Zhang, Tayyab Rehman, Jianxion Han, Sabir Khan, Muhammad Shafiq, Xuegang Yang, Zhongkang Yan, Xingyuan Yang

**Affiliations:** 1Faculty of Health Sciences, Institute of Physical Sciences and Information Technology, Anhui University, 111 Jiulong Road, Hefei 230601, China; bilal.microbiologist@yahoo.com (H.B.); zhang695904964@163.com (G.Z.); q20101013@stu.ahu.edu.cn (J.H.); sabir_khan182@yahoo.com (S.K.); yangxuegang1208@163.com (X.Y.); yzk@stu.ahu.edu.cn (Z.Y.); 2Institute of Basic Medical Sciences, Khyber Medical University, Phase V, Hayatabad, Peshawar 25120, Khyber Pakhtunkhwa, Pakistan; tayyab.ibms@kmu.edu.pk; 3Department of Cell Biology and Genetics, Shantou University Medical College, Shantou 515041, China; drshafiqnjau@yahoo.com

**Keywords:** *K. pneumoniae*, carbapenem resistance, New Delhi metallo-β-lactamase, extended-spectrum-β-lactamases, IncX3 plasmid

## Abstract

The New Delhi Metallo-β-lactamase (NDM) is among the most threatening forms of carbapenemases produced by *K. pneumoniae*, well-known to cause severe worldwide infections. The molecular epidemiology of *bla*_NDM-1_-harboring *K. pneumoniae* is not well elucidated in Pakistan. Herein, we aim to determine the antibiotics-resistance profile, genes type, molecular type, and plasmid analysis of 125 clinically isolated *K. pneumoniae* strains from urine samples during July 2018 to January 2019 in Pakistan. A total of 34 (27.2%) *K. pneumoniae* isolates were carbapenemases producers, and 23 (18.4%) harbored the *bla*_NDM-1_ gene. The other carbapenemases encoding genes, i.e., *bla*_IMP-1_ (7.2%), *bla*_VIM-1_ (3.2%), and *bla*_OXA-48_ (2.4%) were also detected. The Multi Locus Sequence Typing (MLST) results revealed that all *bla*_NDM-1_-harboring isolates were ST11. The other sequence types detected were ST1, ST37, and ST105. The cluster analysis of Xbal Pulsed Field Gel Electrophoresis (PFGE) revealed variation amongst the clusters of the identical sequence type isolates. The *bla*_NDM-1_ gene in all of the isolates was located on a 45-kb IncX3 plasmid, successfully transconjugated. For the first time, *bla*_NDM-1_-bearing IncX3 plasmids were identified from Pakistan, and this might be a new primary vehicle for disseminating *bla*_NDM-1_ in Enterobacteriaceae as it has a high rate of transferability.

## 1. Introduction

The carbapenemases-producing *K. pneumoniae* (CP-KP) are among the deadliest pathogens, producing numerous carbapenem hydrolyzing enzymes. The New Delhi Metallo-β-lactamase (NDM), which belongs to Ambler class B Metallo-β-lactamases (MBL) carbapenemases, degrades carbapenem, penicillin, and cephalosporin and is not inhibited by any β-lactam inhibitors [1]. The *bla*_NDM-1_ was first discovered in *K. pneumoniae* isolated from a Swedish patient having Urinary Tract Infection (UTI) formerly hospitalized in India (New Delhi) [2]. Until now, 21 variants of the NDM have been identified in different regions of the globe [3]. About 51% of all NDMs in the world are reported from China and India, the most populous countries in Asia, and therefore Asia is considered an NDM pool. Several pathogens, including prominent ones like *A. baumannii, E. coli, Pseudomonas* spp, and *K. pneumoniae*, are reported to be NDM producers [4]. The *bla*_NDM-1_-producing pathogenic bacteria disseminate through body fluids and food chains in hospitals and communities [5].

The *bla*_NDM-1_ has been detected on both chromosomal and plasmid DNA. However, its presence on plasmids is more threatening due to its horizontal transferring [6]. Initially, it was found on a 180-kb plasmids; however, later on, it was also reported on 50–500-kb-sized plasmids [2,7]. Among the plasmid incompatibility types, the IncF type is the most prevalent [2]. The Tn125 composite transposon containing *bla*_NDM-1_ and downstream *ble*MBL, along with several other genes set by *ISAba125* elements, is responsible for the global dissemination of the *bla*_NDM-1_ gene. The intact transposon is perceived by *Acinetobacter* spp, while in Enterobacteriaceae different variants truncated on 3’ ends are observed [8].

In Pakistan, the dissemination of NDM is a matter of grave concern. Due to the misuse of antibiotics and no tendency to detect antibiotic resistance’s molecular basis, the resistance genes can easily flow within the community and even produce the deadliest variants. A study from Karachi reported that 57–60% of adult and neonatal deaths in Pakistan occurred due to NDM-producing bacteria [9]. To combat these superbugs, continuous surveillance studies and molecular typing of the bacterial strains and plasmid types are required to depict NDM-producing bacteria. Therefore, the current research, an effort to contribute to the aforementioned concern, investigates CP-KP strains (particularly NDM positive) in a tertiary care hospital located in Islamabad, Pakistan. We have determined the antibiotic-resistance profile, ESBL and carbapenemases encoding genes, molecular typing of the strains, and plasmid analysis of *bla*_NDM-1_ harboring isolates.

## 2. Material and Methods

### 2.1. Samples Collection and Identification

This study was approved by the Ethical Review Committee of Anhui University, Hefei, China (letter No = 2020KY NO. 19). The initial samples collection and processing were performed in Microbiology Laboratory of Pakistan Institute of Medical Sciences (PIMS), Islamabad and the Institute of Basic Medical Sciences, Khyber Medical University, Peshawar, Pakistan. Molecular investigations were carried out at Yang lab (C312), Institute of Physical Sciences and Information Technology, Anhui University, Hefei, China. The standard microbiological aseptic techniques were used in the processing and transportations of samples [10].

A total of 230 urine samples were collected from a tertiary care hospital, PIMS, located in Pakistan’s capital city, Islamabad, from July 2018 to January 2019. The samples were collected in sterile urine sample collection bottles and immediately transferred to the Microbiology Laboratory of PIMS hospital. Initially, the samples were directly streaked on Cysteine–lactose–electrolyte-deficient agar (CLED) and incubated at 37 °C for 18–24 h. The growths of 125 isolated *K. pneumoniae* were further evaluated on HiChrom *Klebsiella* selective agar (HiMedia^®^) at 37 °C for 24 h. The isolates were preserved in the Luria-Bertani medium (LB) supplemented with 40% glycerol at −80 °C until further processing. For antibiotic susceptibility testing and other molecular investigations, the working cultures were maintained on nutrient agar and LB media at 2–8 °C for up to four weeks. Molecular identification was performed by amplification of 16S rDNA using specific primers (Appendix A). The sequencing results were analyzed via EZ Bio-cloud software (https://www.ezbiocloud.net/, accessed on 26 April 2021).

### 2.2. Antibiotic Susceptibility Testing

The antibiotic susceptibility testing of all the isolates was performed using the broth microdilution method according to the ISO standard 20776–1 [11]. All the antibiotics were purchased from Sigma-Aldrich (Shanghai, China) Trading Co., Ltd. Nine different antibiotics, amikacin (CAS No: 37517-28-5), ceftriaxone (CAS No: 104376-79-6), cefotaxime (CAS No: 64485-93-4), ceftazidime (CAS No: 120618-65-7), meropenem (CAS No: 119478-56-7), imipenem (CAS No: 74431-23-5), ciprofloxacin (CAS No: 85721-33-1), tetracycline (CAS No: 64-75-5), and tobramycin (CAS No: 32986-56-4) activities, were evaluated. The susceptibility testing was performed in duplicate for each antibiotic. The *E. coli* ATCC 25922 was used as negative control for the susceptibility testing of all antibiotics, and *K. pneumoniae* ATCC1705 were used as positive control for the susceptibility testing of carbapenem antibiotics. The results were interpreted according to the Clinical & Laboratory Standards Institute (CLSI) 2020 [12].

### 2.3. Screening of Carbapenemases and ESBL

The carbapenemases-producing abilities were determined by the Carba NP test and Modified Hodge Test (MHT). Initially, the MHT was performed to detect the carbapenemases-producing ability of the isolates. Briefly, the tenfold dilution of 0.5 McFarland *E. coli* ATCC 25,922 (carbapenem susceptible indicator strains) culture lawn was made on Muller Hinton Agar (MHA), and a 10-µg ertapenem disc was placed in the center of the plate. The testing isolate was lined from the disc up to the edge of the petri dish. *K. pneumoniae* ATCC1705 (MHT Positive) and *K. pneumoniae* ATCC1706 (MHT Negative) were used as a control recommended by CLSI 2020. The plates were incubated for 18–24 h at 37 °C. The confirmation of MHT positive isolates was done by the Carba NP test recommended by the CLSI 2020. The ESBL activity of Carba NP positive isolates was also detected via the double-disk synergy test (DDST) following CLSI guidelines [12].

### 2.4. Detection of Antibiotic Resistance Genes

The Carba NP positive isolates were subjected to an amplification of the antibiotic resistance gene. The whole genomic DNA of the positive isolates was extracted using the conventional boiling method [13]. The quantity and quality of extracted DNA were analyzed by one drop^TM^ (OD-1000+, Spectrophotometer). The primers specific for resistant genes i.e., *bla*_NDM-1_, *bla*_KPC-2_, *bla*_VIM_, *bla*_IMP_, *bla*_GES_, *bla*_OXA-48_, *bla*_TEM_, *bla*_SHV_, *bla*_CTXM_, and *bla*_OXA_, as mentioned in Appendix A, were used for amplification of the resistance genes. The amplified genes were run on 1.5% agarose gel and were subsequently sequenced. For confirmation of the resistant gene and variant types, the sequence results were analyzed via the NCBI blast tool (https://blast.ncbi.nlm.nih.gov/Blast.cgi, accessed on 26 April 2021).

### 2.5. Multi Locus Sequence Typing (MLST)

MLST was used to detect the sequence types of carbapenemases-producing isolates. The sequence outcome of seven alleles, *rpoB*, *gapA, mdh*, *pgi*, *phoE*, *infB*, and *tonB*, were determined from the Pasteur MLST database (https://bigsdb.pasteur.fr/klebsiella/klebsiella.html, accessed on 26 April 2021). The phylogenetic analyses based on MLST alleles profile were carried out on BioNumerics (Applied Maths, Sint-Martens-Latem, Belgium).

### 2.6. Restriction Enzyme Analysis with Pulsed Field Gel Electrophoresis (REA-PFGE)

Furthermore, the Xbal PFGE was carried out according to the PFGE protocol to determine the genetic relatedness between CP-KP strains [14]. DNA cluster analysis was performed by BioNumerics v.8.0 (Applied Maths, Sint-Martens-Latem, Belgium). The Dice similarity coefficient for cluster analysis was calculated with a position tolerance of 1.5%, and a dendrogram was created based on the unweight pair-group method with arithmetic mean (UPGMA).

### 2.7. Transconjugation

The transconjugation experiment was carried out to determine the transferability of *bla*_NDM-1_ positive isolates. The *bla*_NDM-1_ harboring *K. pneumoniae* strains were taken as the donors, and *E. coli* EC 600 (Nal^R^, Rif^R^) were chosen as recipients. The rifampicin 600 mg/L and meropenem 4 mg/L were used to select transconjugants. The experiment was performed as designed previously [15]. The transconjugants were evaluated from their antibiotic sensitivity testing and *bla*_NDM-1_ gene PCR, as described previously.

### 2.8. Plasmids Analysis

The PCR-based replicon typing (PBRT) of *bla*_NDM-1_-harboring transconjugants *E. coli* EC 600 (Nal^R^, Rif^R^) was performed to investigate the plasmid incompatibility type carrying the *bla*_NDM-1_ gene. The alkaline lysis method was used for the extraction of plasmid DNA from transconjugants [16]. The PBRT 2.0 kit (MBK0078, Diatheva, Cartoceto, Italy) was used to detect the plasmid type. The expected band size of eight multiplex PCR was run on 2% agarose gel, and the results were analyzed following the kit guidelines.

S1 PFGE and Southern blotting were conducted to determine the sizes of *bla*_NDM-1_-harboring plasmids. Five randomly selected *bla*_NDM-1_-harboring transconjugants *E. coli* EC 600 (Nal^R^, Rif^R^) strains were taken for this experiment. S1 PFGE was performed by following the previously described protocol [17]. The clamped homogeneous electric fields (CHEF) mapper PFGE (Bio-Rad, Hercules, CA, USA) was set for the experiment as a 21-h run time, initial switch time of 2.3 s, and final switch time of 1 min 20 s at 6 V/cm. The capillary transfer technique was performed to transfer the plasmids from the gel to nylon membrane [18]. According to the kit guidelines (Roche Diagnostics, Mannheim, Germany), Southern blotting was carried out using the digoxin-labeled *bla*_NDM-1_-specific probe.

## 3. Results

### 3.1. Bacterial Isolates and Antibiotic Sensitivity Profile

A total of 125 *K. pneumoniae* isolates were confirmed via 16S rDNA from urine samples (n = 230) to detect carbapenem-resistant *K. pneumoniae* in Pakistan’s tertiary care hospitals. On the basis of 16S rDNA sequencing, we confirmed that all the isolates were *K. pneumoniae* subsp. *pneumoniae*. The antibiotic susceptibility testing of all the isolates was performed. Cefotaxime and tetracycline were found to be the most resistant drugs, having a resistivity of 80% and 71%, respectively. Imipenem was found to be the most susceptible drug (11%) among the nine tested antibiotics. The complete profile of all the isolates for nine tested antibiotics is depicted in Figure 1.

### 3.2. Carbapenemases Producing Isolates

To find out the carbapenemases-producing ability, the MHT and Carba NP were performed. Among 125 *K. pneumoniae* isolates, 34 strains (27.2%) were MHT- and Carba NP-positive for carbapenemases production. The ESBL-producing ability in carbapenemases-producing isolates was carried out via DDST. All of the carbapenemases-producing isolates were also ESBL-positive.

### 3.3. Antibiotic-Resistant Genes

The genes encoding carbapenemases were determined using conventional PCR with gene-specific primers. The results showed that *bla*_NDM-1_ was present in 23 (18.4%), *bla*_IMP-1_ in nine (7.2%), *bla*_VIM-1_ in four (3.2%), and *bla*_OXA-48_ in three (2.4%) isolates, while *bla*_KPC-2_ and *bla*_GES_ were not detected. The ESBL-encoding genes were also investigated in CP-KP isolates. Co-occurring carbapenemases- and ESBL-encoding genes were detected in theses isolates. The coexisting genes of *bla*_NDM-1_ and *bla*_CTXM-15_ were 11 (8.8%), of *bla*_NDM-1,_
*bla*_IMP-1,_ and *bla*_CTXM-15_ were three (2.4%), of *bla*_NDM-1,_
*bla*_SHV-75_, and *bla*_CTXM-15_ were three (2.4%), of *bla*_NDM-1,_
*bla*_OXA-48,_ and *bla*_TEM-1_ were one (0.8%), of *bla*_NDM-1,_
*bla*_OXA-48,_ and *bla*_SHV-75_ were one (0.8%), of *bla*_IMP-1_ and *bla*_OXA-1_ were four (3.2%), of *bla*_IMP-1_ and *bla*_TEM-1_ were two (1.6%), and of *bla*_VIM-1_ and *bla*_CTXM-15_ were two (1.6%). The percentage of ESBLs encoding genes in co-existence with a particular carbapenem resistance genes are presented in (Table 1).

### 3.4. MLST

The Pasteur MLST database protocol was followed to find out the sequence type of CP-KP isolates. The MLST results revealed that 27 isolates were ST11, three were ST1, two were ST37, and two others were ST105. Among these, the 23 *bla*_NDM-1_-harboring isolates were ST11. Similarly, the three *bla*_IMP-1_- and one *bla*_VIM-1_-carrying isolates were also ST11. Among the three ST1 strains, two were *bla*_IMP-1_-harboring isolates and one was a *bla*_VIM-1_-harboring isolate. Among the two ST37 strains, one was *bla*_IMP-1_ and one was *bla*_VIM-1_. Likewise, among the two ST105 strains, one was a *bla*_OXA-4_ - and one was a *bla*_VIM-1_-bearing isolate. The clonality of carbapenemases-producing isolates was evaluated as a minimum spanning tree (MST) based on the MLST alleles profile (Figure 2).

### 3.5. REA-PFGE

The molecular typing of *K. pneumoniae* isolates was performed by using the Xbal PFGE technique. After successfully restricting endonucleases of CP-KP isolates via Xbal enzymes, the DNA clusters were analyzed using BioNumerics software. The cluster analysis of Xbal PFGE revealed a variation amongst the clusters of the identical sequence types’ isolates (Figure 3).

### 3.6. Transconjugation

The transconjugation experiment assessed the transferability of *bla*_NDM-1_ from the donors’ *bla*_NDM-1_-harboring strains to *E. coli* EC 600 (Nal^R^, Rif^R^). All of the *bla*_NDM-1_-carrying strains were successfully transconjugated. The results showed that the bacterial colonies that appeared on a selective plate having 4 µg/mL meropenem and 600 µg/mL rifampicin were considered transconjugants. The colonies were confirmed via antimicrobial susceptibility testing and PCR for the *bla*_NDM-1_ gene. All the isolates harbored the *bla*_NDM-1_ gene, and their MIC for meropenem was 128 µg/mL. This indicates the *bla*_NDM-1_ presence on the plasmid.

### 3.7. Plasmid Analysis

To find out the plasmid incompatibility type of *bla*_NDM-1_-harboring isolates, PBRT was performed. The PBRT result of *bla*_NDM-1_-harboring transconjugants revealed the presence of only the IncX3 plasmid in all of the isolates. To further determine the size of the *bla*_NDM-1_-harboring plasmid, S1 PFGE and Southern blotting were carried out. The results revealed the presence of a 45-kb IncX3 plasmid carrying the *bla*_NDM-1_ gene in all of the isolates (Figure 4).

## 4. Discussion

The *K. pneumoniae*, a most important global cause and vehicle for spreading antibiotic resistance, has been declared a global health threat by The United State Centers for Disease Control and Prevention. The carbapenem resistance, a type of β-lactamase resistance in MDR *K. pneumoniae,* is one of the most alarming types, commonly known as CP-KP [19]. The Indo-Pak subcontinent is considered a reservoir of CP-KP. However, minimal numbers of studies are available from Pakistan on this matter [20]. In the present study, we aimed to identify the carbapenemases-producing *K. pneumoniae* isolates from urine samples in Pakistan’s tertiary care hospitals. We identified CP-KP in 34 isolates out of 125 *K. pneumoniae* from urine samples. The *bla*_NDM-1_ gene was detected in 23 out of 34 CP-KP isolates. Some other clinically significant antibiotic resistance genes like *bla*_OXA-48,_
*bla*_IMP-1_, *bla*_VIM-1_, *bla*_CTXM-15,_
*bla*_TEM-1,_ and *bla*_SHV-75_ were also detected in the present study. The prevalence of *bla*_NDM-1_ was 95.83% in CP-KP isolates, while it was calculated as being 18.4% in all tested *K. pneumoniae* isolates. The results revealed that *bla*_NDM-1_ producers are among the severe problems in CP-KP isolates. In Pakistan, the coharboring of carbapenem-resistant genes is not often noticed in *K. pneumoniae*. Even so, in a study, the coexistence of *bla*_NDM-1_ with other ESBL genes was identified [21]. In 2019, a study from Peshawar, Pakistan resembled our findings by reporting a 16% *bla*NDM-1 prevalence in Gram-negative bacteria [22]. Nevertheless, some other studies have reported different patterns of prevalence for *bla*_NDM-1_ in Pakistan, i.e., 41% in 2013 and 6.8% in 2018 [23,24]. The differences between the results might be due to a minimal number of CP-KP studies in human isolates from Pakistan. Moreover, the *bla*_NDM-1_ prevalence also varies with different sample sources [20], as our samples are solely from urine origin, while the abovementioned results have been published from multiple sample sources.

The MLST result revealed that all of the *bla*_NDM-1_-harboring isolates were ST11. The ST11 is a single locus variant of ST258 and a worldwide known high-risk clone for carbapenemases dissemination [25,26]. The presence of *bla*_NDM-1_ on ST11 has been previously reported in various world regions, i.e., India, China, Balkan, Bulgaria, and many others [26,27]. The ST11 is classically related to multidrug resistance attainment due to its tendency for harboring multiple plasmids and multiple resistance genes [25,28]. The ST11 *K. pneumoniae* coharboring *bla*_NDM-1_ and ESBL genes has been identified in Iran and the United Arab Emirates [29,30]. In Pakistan, a 2018 study reported ST11 CP-KP with coexisting *bla*_NDM-1_ and ESBL genes [31]. The result of the current study and available literatures suggest that ST11 subclones with various β-lactamases have an increased epidemic potential.

The plasmid incompatibility type of *bla*_NDM-1_-harboring *K. pneumoniae* in the present study is IncX3 having horizontal transferability. To the best of our knowledge, this is the first study from CP-KP clinical isolates in Pakistan that has *bla*_NDM-1_-bearing IncX3 plasmids. Only one study from Pakistan demonstrated the MBL gene, i.e., NDM-7 occurrence on IncX3 plasmids in *E. coli* isolated from Chicken meat samples [32]. However, IncX3 bearing NDM genes are prevalent in other regions of the world, have a limited host range, and are mainly observed in Enterobacteriaceae [33,34]. Nevertheless, their carriage rate of *bla*_NDM_ is higher among the other carbapenem-resistant genes [35]. The finding of NDM-bearing IncX3 in *K. pneumoniae* in the present study and *E. coli* in other studies from Pakistan suggests the distinctive role of IncX3 plasmids in intergeneric *bla*_NDM-1_ transmission [32].

## 5. Conclusions

In the present study, a *bla*_NDM-1_-harboring IncX3 plasmid is identified for the first time in clinically isolated *K. pneumoniae* from urine samples in Pakistan. The finding suggests that the IncX3 plasmid might be a new primary vehicle in Pakistan for disseminating *bla*_NDM-1_ in Enterobacteriaceae, as it has a high transferability. Urgent and large-scale surveillance studies targeting *bla*_NDM-1_ and IncX3 plasmids are required to depict the current scenario. Moreover, the initiation of infection control commissions in hospitals as well as the execution of accurate screenings, Carba NP tests, and complete isolation are required to control the spread of resistant isolates.

## Figures and Tables

**Figure 1 microorganisms-09-00951-f001:**
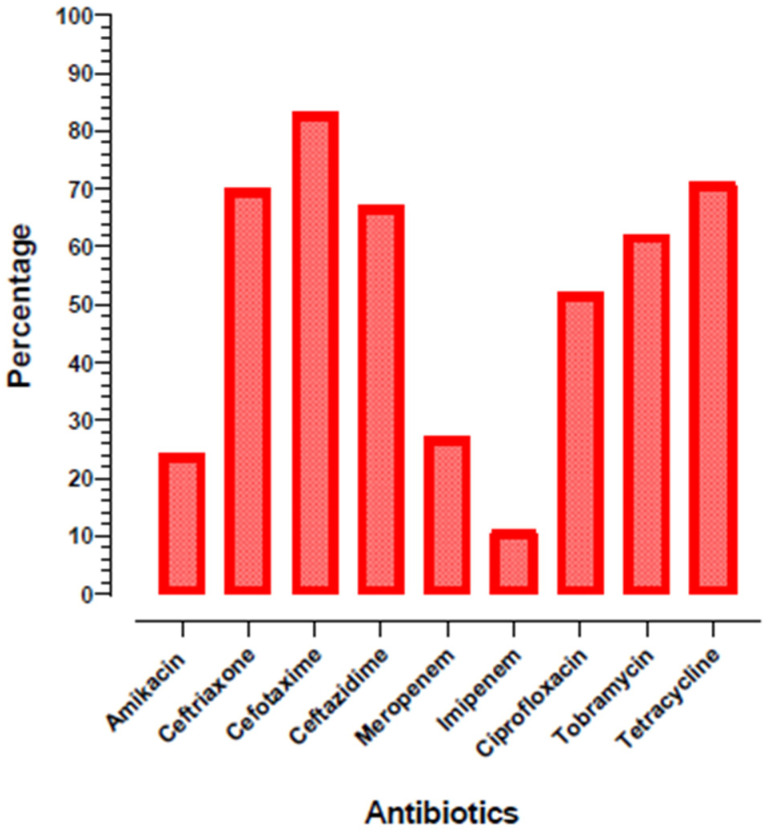
The percentage of antibiotic resistance profile of nine antibiotics against 125 *K. pneumoniae* isolates.

**Figure 2 microorganisms-09-00951-f002:**
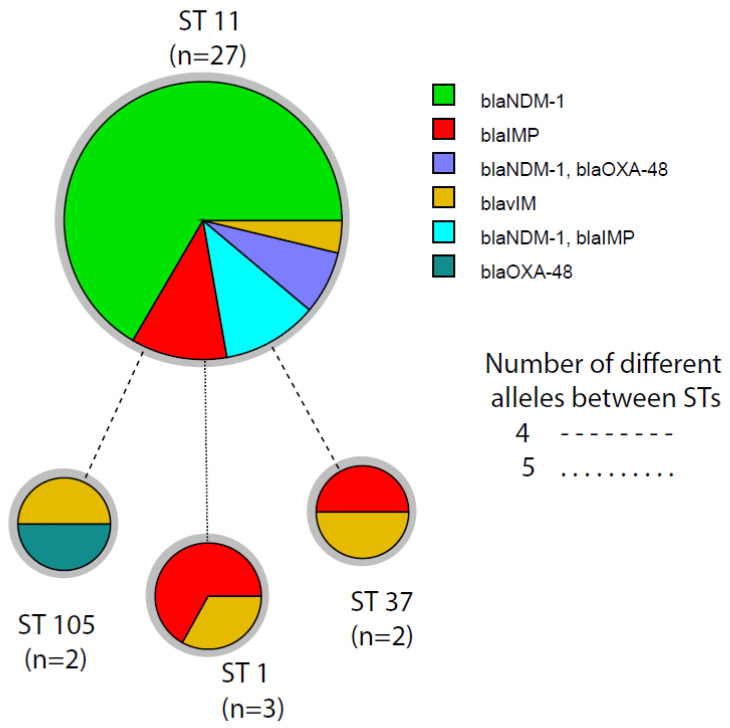
Minimum spinning tree of carbapenemases-producing *K. pneumoniae* isolates based on the MLST alleles and sequence type (ST). The nodes represent the STs, sizes of nodes characterize the number of isolates, and length of branches shows the number of different alleles out of seven MLST alleles. Nodes are labeled with the corresponding sequence type.

**Figure 3 microorganisms-09-00951-f003:**
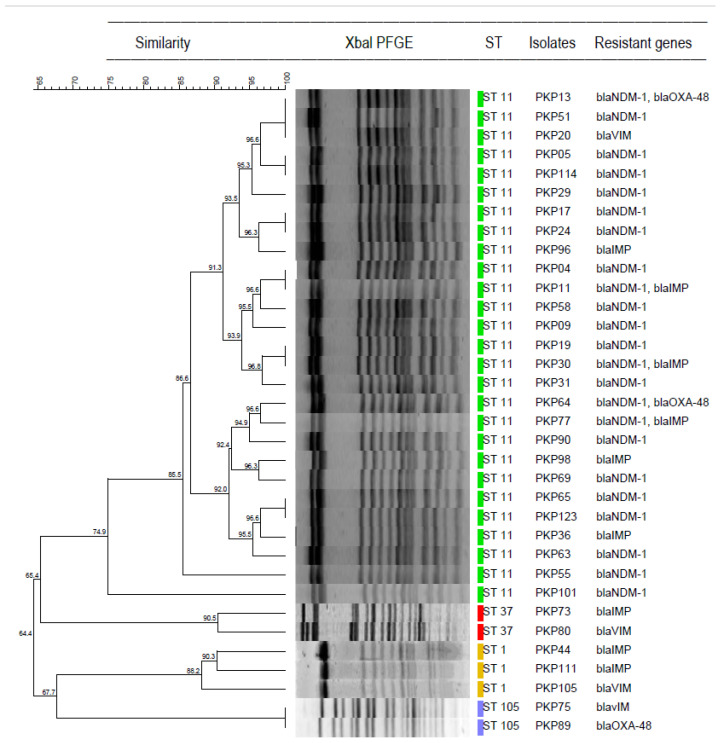
Xbal PFGE result of carbapenemases-producing *K. pneumoniae*. The cluster analysis and dendrogram is constructed by BioNumerics V 8.0. ST is short for sequence type.

**Figure 4 microorganisms-09-00951-f004:**
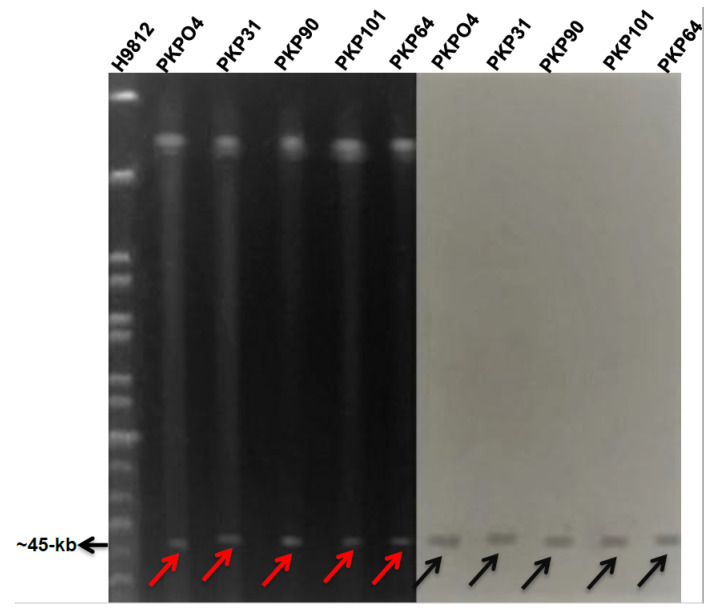
S1 PFGE and Southern blot results. PKPO4, PKP31, PKP90, PKP101, and PKP64 are the isolates, and H9812 is the molecular size marker (1135 to 20.5 kb) digested with Xbal enzymes. Southern hybridization revealed the *bla*_NDM-1_ probed on the 45-kb IncX3 plasmid. Black arrows show *bla*_NDM-1_-bearing plasmids on the gel, while red arrows represent *bla*_NDM-1_-harboring plasmids on the nylon membrane.

**Table 1 microorganisms-09-00951-t001:** Prevalence of antibiotic resistance genes (alone or in coexistence) investigated in the present study.

Carbapenem Resistance Genes	N# (%)	ESBLs Encoding Genes	N * (%)
*bla* _NDM-1_	23 (18.4%)	*bla* _CTXM-15_	17 (13.6%)
		*bla* _TEM-1_	1 (0.8%)
		*bla* _SHV-75_	4 (3.2%)
*bla* _IMP-1_	9 (7.2%)	*bla* _CTXM-15_	3 (2.4%)
		*bla* _OXA-1_	4 (3.2%)
		*bla* _TEM-1_	2 (1.6%)
*bla* _VIM-1_	4 (3.2%)	*bla* _CTXM-15_	2 (1.6%)
*bla* _OXA-48_	3 (2.4%)	*-*	-

Footnote: N# = number of isolates having particular carbapenem-resistant genes out of 125 isolates. N * = number of isolates showing the coexistence of carbapenem-resistant genes and ESBL-encoding gene out of 125 isolates. Note: The sum of percentages is not equal to 100 because most of the isolates harbor multiples genes, with each gene being counted separately.

## Data Availability

All the data is presented in the manuscript.

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
