# Peer review of "First Report of blaNDM-1 Bearing IncX3 Plasmid in Clinically Isolated ST11 Klebsiella pneumoniae from Pakistan"

_microorganisms, 2021, doi:10.3390/microorganisms9050951_

Round 1
Reviewer 1 Report
Review Report (Manuscript ID microorganisms-1193998)
The topic of this manuscript falls within the scope of Microorganisms (MDPI). The authors performed a very important and current problem of nosocomial infections by Klebsiella pneumoniae NDM-1. Klebsiella pneumoniae can be a physiological component of the microbiota digestive tract, but Kp can also colonizing skin and mouth, especially among the medical Staff, these bacteria spread freely in a hospital environment, and can infect immunocompromised patients causing incl., UTI, pneumonia, sepsis, soft tissue infections, and inflammation peritoneum. K.pneumoniae strains acquire genes encoding virulence factors and drug resistance, e.g., by horizontal gene transfer (HGT). The most dangerous from the point of view of therapy is a resistance mechanism NDM-1 (New Delhi Metallo-beta-lactamase- 1). The authors of this manuscript describe an essential problem of K. pneumoniae strains isolated in a Pakistani hospital. They determine genes related to resistance, sequence type by MLST and pattern of REA-PFGE isolates in epidemiological research, the authors detected IncX3 plasmid responsible for carbapenem resistance.
My comments on the manuscript
Dissemination of blaNDM-1 bearing IncX3 Plasmid in clinically isolated ST11 Klebsiella Pneumoniae from Pakistan.
The title should be without a dot.
Line 3: the second part of the species name: Klebsiella Pneumoniae to Klebsiella pneumoniae
Introduction: Line52: The tn125 composite transposon containing blaNDM-….. should be: capital letter Tn125
Methods: The experiments were planned and performed correctly, using the typical methods for this type of research; no information about the consent of the bioethics committee
After developing a classification system based on 16S rDNA analysis, five clusters were defined with the assigned species. Cluster V of Klebsiella pneumoniae spp. pneumoniae, Klebsiella pneumoniae spp. rhinoscleromatis and Klebsiella pneumoniae spp. ozaenae. The authors also confirmed the species by amplification of 16S rDNA and sequencing, hence they should have information about spp. Was it Klebsiella pneumoniae spp. pneumoniae? (Klebsiella pneumoniae sensu stricte)? What was the purpose of the sequencing? It is not required in hospital practice.
Molecular typing by MLST and REA-PFGE methods should be in separate chapters, each of these methods has a different purpose (MLST for global information; REA-PFGE for local information).
Line 143 should be: capital letter - Southern blotting
Results
Table 1 should be in a different layout
Conclusion
Line 268: typos - should be K.pneumoniae
References No self-citations
Recommendation I think this paper is well written, readable and informative. I recommend it for publication after minor revision.
Reviewer 2 Report
Bilal et al. conducted a surveillance study and demonstrated the presence and molecular characterization of carbapenem resistant K. pneumoniae strains in Pakistan. The study is well designed and the manuscript is well written. I have some comments as described below.
Specific comments:
Line 18-20: “The genes investigation, molecular typing, and plasmid analysis of carbapenem-resistant K. pneumoniae isolates were performed”. This sentence seems redundant to the line 14-15.
Line 20: “About 34”. Is it an approximate number or absolute number of isolates that were carbapenemase producers?
Line 70-78: This section needs more clarification on-
- How the specimens were collected and transported to the laboratory?
- What was the growth condition?
Line 80-84: This section needs more detail description on-
- which broth microdilution method was followed and requires a citation (i.e., is it ISO 20776-1 or other standard?).
- Detail manufacturer information is required for all the antimicrobial agents.
- How the quality control was checked for BMD? What positive and negative control strains were used?
- A citation is required for CLSI 2020 standards.
Line 89: “ATCC 25922”?
Line 94: “Clinical & Laboratory Standards Institute (CLSI)” This elaboration should appear earlier in the manuscript once.
Line 146: From how many urine samples? This number should be mentioned here.
Line 154: Figure 1. The figure shows redundant data. Either resistance or susceptible percentage would be sufficient to describe this data. Don’t need to present both resistance and susceptible percentage.
